# Combination of chemotherapy and PD-1 blockade induces T cell responses to tumor non-mutated neoantigens

Alessio Grimaldi[1,14], Ilenia Cammarata[1,14], Carmela Martire[1], Chiara Focaccetti [1], Silvia Piconese[1], Marta Buccilli[1], Carmine Mancone[2], Federica Buzzacchino[3], Julio Rodrigo Giron Berrios[3], Nicoletta D'Alessandris[4], Silverio Tomao[3], Felice Giangaspero[4,5], Marino Paroli[6], Rosalba Caccavale[6], Gian Paolo Spinelli[7], Gabriella Girelli[2], Giovanna Peruzzi [8], Paola Nisticò [9], Sheila Spada[9], Mariangela Panetta[9], Fabiana Letizia Cecere[10], Paolo Visca[11], Francesco Facciolo[12], Flavia Longo[3] & Vincenzo Barnaba[1,8,13 ✉]

Here, we developed an unbiased, functional target-discovery platform to identify immunogenic proteins from primary non-small cell lung cancer (NSCLC) cells that had been induced to apoptosis by cisplatin (CDDP) treatment in vitro, as compared with their live counterparts. Among the multitude of proteins identified, some of them were represented as fragmented proteins in apoptotic tumor cells, and acted as non-mutated neoantigens (NM-neoAgs). Indeed, only the fragmented proteins elicited effective multi-specific CD4$^+$ and CD8$^+$ T cell responses, upon a chemotherapy protocol including CDDP. Importantly, these responses further increased upon anti-PD-1 therapy, and correlated with patients' survival and decreased PD-1 expression. Cross-presentation assays showed that NM-neoAgs were unveiled in apoptotic tumor cells as the result of caspase-dependent proteolytic activity of cellular proteins. Our study demonstrates that apoptotic tumor cells generate a repertoire of immunogenic NM-neoAgs that could be potentially used for developing effective T cell-based immunotherapy across multiple cancer patients.

---

[1] Dipartimento di Medicina Interna e Specialità Mediche, Sapienza Università di Roma, 00161 Rome, Italy. [2] Dipartimento di Medicina Molecolare, Sapienza Università di Roma, 00161 Rome, Italy. [3] Dipartimento di Scienze Radiologiche, Oncologiche e Anatomo Patologiche, Oncologia Medica, Università di Roma, 00161 Rome, Italy. [4] Department of Radiological, Oncological and Pathological Sciences, Sapienza University of Rome, Rome, Italy. [5] IRCCS Neuromed, Pozzilli, Isernia, Italy. [6] Dipartimento di Scienze e Biotecnologie Medico-Chirurgiche, Sapienza Università di Roma - Polo Pontino, 04100 Latina, Italy. [7] UOC Oncologia Universitaria, ASL Latina (distretto Aprilia), Sapienza Università di Roma, Via Giustiniano snc, 04011 Aprilia, Latina, Italy. [8] Center for Life Nano Science@Sapienza, Istituto Italiano di Tecnologia, 00161 Rome, Italy. [9] Tumor Immunology and Immunotherapy Unit, IRCCS-Regina Elena National Cancer Institute, Rome, Italy. [10] Medical Oncology 1, IRCCS-Regina Elena National Cancer Institute, Rome, Italy. [11] Unit of Pathology, IRCCS-Regina Elena National Cancer Institute, Rome, Italy. [12] Thoracic Surgery Unit, IRCCS-Regina Elena National Cancer Institute, Rome, Italy. [13] Istituto Pasteur - Fondazione Cenci Bolognetti, 00185 Rome, Italy. [14] These authors contributed equally: Alessio Grimaldi, Ilenia Cammarata. ✉email: vincenzo.barnaba@uniroma1.it

   1

Lung cancer is the major cause of cancer-related death worldwide, and non-small-cell lung cancer (NSCLC) accounts for 85% of lung cancer cases, of which lung adenocarcinoma (LUAD) and lung squamous cell carcinoma (LUSC) are the most common subtypes[1]. Additionally, NSCLC is molecularly heterogeneous, because of the amazing diversity of somatic genome mutations, including mutations of genes contributing to carcinogenesis (driver genes; e.g., *KRAS*, *EGFR*, *TP53*, etc.)[2], and passenger mutations of genes unrelated to tumor growth, a large part of which are patient specific (private) and confers a huge antigenic heterogeneity to various tumors including NSCLC[3,4].

In the past decade, immunotherapy has achieved extraordinary objective response rates for the treatment of tumors, particularly by using monoclonal antibodies (mAbs) specific to various inhibitory signaling molecules (immune checkpoints, such as programmed death-1 [PD-1] or cytotoxic T-lymphocyte antigen-4 [CTLA-4]) that are strongly upregulated by exhausted tumor-infiltrating lymphocytes[5–9]. These mAbs (defined as checkpoint inhibitors) unleash anti-tumor T cells, leading to a reliable shrinkage of several metastatic tumors. However, checkpoint inhibitors currently used to target CD8$^+$ T cells (e.g., anti-CTLA-4 and/or anti-PD-1 mAbs) have not been shown to be efficient for all tumor types and may cause a partial remission in the majority of tumors. Innovative immunotherapy strategies are in progress, particularly based on the combination of different approaches, including new and current inhibitors providing immune checkpoint blockade (ICB), as well as tumor vaccines against neoantigens (neoAgs) that are generated as a consequence of the wide tumor genome mutations[4]. T cells specific to either public or private neoAgs, which are encoded by somatic gene mutations, are not purged by central tolerance[10], can migrate in the periphery, and be of particular relevance to tumor control[4,11–18]. This conclusion is supported by the association between tumor T cell infiltration and mutational load, or between the expansion of mutated neoAg-specific T cells and the reinvigoration of anti-tumor T cell immunity following ICB[14,17,19]. However, despite the high mutational burden and the huge tumor T cell infiltration in several tumors including NSCLC, the frequencies of the related neoAg-specific T cells are relatively very low[20–23]. In the light of these evidences, it is possible that a consistent proportion of tumor-associated T cells may be specific to non-mutated (NM)-neoAgs generated by various forms of protein modifications occurring at post-transcriptional level, such as protein splicing[24], dysregulated phosphorylation or glycosylation[25–27], proteasome generation of spliced peptides[28], peptide citrullination[29], impaired peptide processing in TAP-deficient tumor cells[30], or proteasomal degradation of defective ribosomal products (DRiPs)[31]. These NM-neoAgs may provide rational targets for cancer immunotherapy, because they should not be expressed or expressed at concentrations that are not enough to delete specific T cells in the thymus.

Potentially, also chemotherapy or radiotherapy may generate immunogenic NM-neoAgs in dying tumor cells, through their capacity to induce immunogenic cell death (ICD)[32–35]. ICD can enable tumor cells to provide both novel antigenic signals, including caspase-cleaved fragmented proteins during apoptosis[36,37], and costimulatory signals, such as a distinct sets of damage-associated molecular patterns (ATP, calreticulin, HMGB1, etc.) principally converting tolerogenic into stimulatory dendritic cells (DCs)[32–35,38–42]. Then, DCs acquire high capability to migrate, to phagocytose dying cells, to process more efficiently caspase-cleaved cellular proteins (i.e., NM-neoAgs), and to cross-prime CD8$^+$ T cells that can provide tumor control[35,38,41,42], on the one hand, and immunopathology in various forms of chronic inflammatory diseases, on the other hand[37,43–48]. In addition, several studies found that ICD synergizes with ICB therapy to further improve T cell responses against different tumors, proposing hence the hypothesis that ICD converts tumor cells into endogenous vaccine and boosts the ICB effects[49–54]. However, despite the large body of evidences on how ICD occurs, few evidences have been reported about the nature of antigens becoming immunogenic upon ICD[41,42].

Here, we used stable isotope labeling by amino acids in cell culture (SILAC)-based mass spectrometry (MS) to quantitatively compare the proteome of primary NSCLC cells that had been made apoptotic by cisplatin (CDDP) treatment in vitro and their live counterparts. The expression of a multitude of proteins was found differently regulated, some of which were represented as fragmented proteins in CDDP-treated apoptotic (CDDP-ap) tumor cells, as the result of a caspase-dependent proteolytic activity. Then, we interrogated memory T cells from NSCLC patients, to detect whether these fragmented proteins were generated in tumor cells following chemotherapy-induced apoptosis, and whether they could act as immunogenic NM-neoAgs capable of inducing potentially protective anti-tumor immune responses.

## Results

### Identification of proteome changes in apoptotic NSCLC cells.
SILAC-based MS was used to identify changes in the proteome of primary NSCLC cells, as previously described[55] (Fig. 1). In brief, primary NSCLC cells (named EpT1Lu line, isolated from a surgery specimen obtained and characterized, as described in Methods section) (Supplementary Fig. 1) were metabolically labeled with heavy ($^{13}C_6$-Lys and $^{13}C_6$-$^{15}N_4$-Arg) isotope medium, and subsequently induced to apoptosis by 0.625 μM CDDP treatment (72 h) (Fig. 1a). Apoptosis was validated by the consistent upregulation of Annexin V and activated caspase-3 in CDDP-treated cells (Supplementary Fig. 1). Non-labeled EpT1Lu cells were grown in light medium ($^{12}C_6$-Lys and $^{12}C_6$-$^{14}N_4$-Arg) and maintained in viable condition. Then, apoptotic cells and live cells were sorted from heavy and light NSCLC populations, respectively (Fig. 1a). A reverse SILAC experiment (SILAC reverse), switching heavy and light media, was also performed. Apoptotic (heavy) and live (light) cells obtained in five independent experiments were pooled and lysed, and equal amounts of proteins from each cell line were mixed (Fig. 1a). One hundred micrograms of this sample was separated by sodium dodecyl sulfate–polyacrylamide gel electrophoresis (SDS-PAGE) and the gel lane was cut into 16 sections ranging from 5 to 250 kDa (Fig. 1b). Proteins from each gel section were digested and quantified using liquid chromatography (nanoLC) followed by 5800 MALDI-TOF/TOF (matrix-assisted laser desorption/ionization tandem time-of-flight) analysis. By this procedure, among the 815 proteins differently expressed between apoptotic (heavy) and live (light) NSCLC cells, some of them resulted as fragmented proteins upregulated in apoptotic NSCLC cells (Fig. 1b). To select fragmented proteins originated by CDDP-dependent apoptosis, we used the electrophoresis-derived molecular weight (MWexp) of the protein as the identification constraint. Fragmented proteins were identified on the basis of the evidence that the MWexp of a polypeptide resulted lower than the theoretical molecular weight (MWcal) of the corresponding entire protein (Fig. 1b and Table 1). Comparison between apoptotic (heavy) and live (light) NSCLC cells in forward SILAC experiments showed that out of a total of 815 proteins identified and quantified (Supplementary Data 1 reported in Description of additional supplementary items), 253 proteins (31%) were found overabundant, 439 (54%) proteins were found downregulated, whereas 16 (2%) resulted fragmented proteins upregulated in apoptotic cells (Fig. 1c). Notably, similar results were found in the reverse SILAC, thus

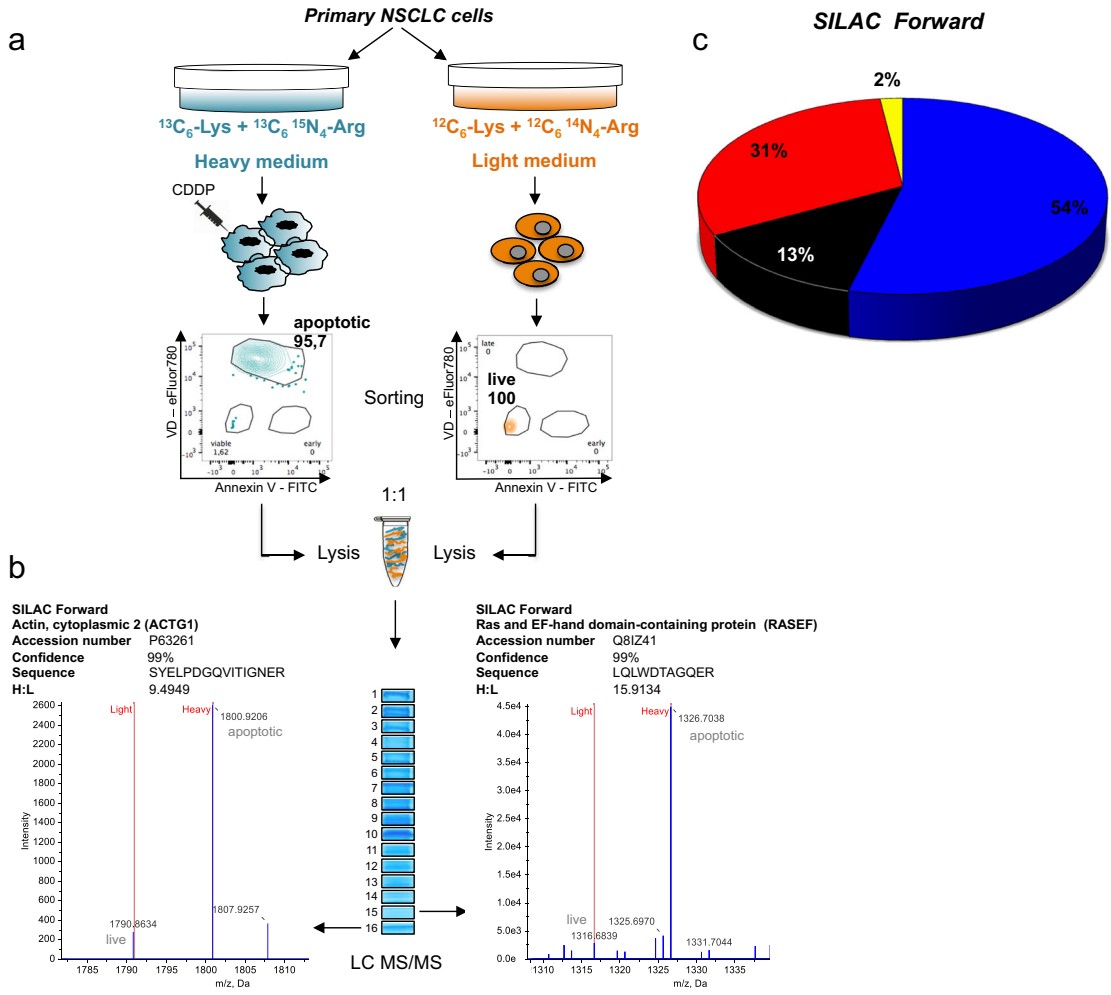

**Fig. 1 Identification of proteome in primary apoptotic and live NSCLC cells by SILAC-based MS. a** Primary NSCLC cells (EpT1Lu) were either labeled with heavy ($^{13}C_6$-Lys and $^{13}C_6$-$^{15}N_4$-Arg) medium or conditioned with light ($^{12}C_6$-Lys and $^{12}C_6$-$^{14}N_4$-Arg) medium. The heavy isotope-labeled cells were induced to apoptosis by 0.625 μM CDDP treatment (apoptotic cells). Apoptotic (heavy) and live (light) fractions were sorted and lysed; equal amounts of protein extracts were mixed in a 1:1 ratio and separated in SDS-PAGE, enzymatically digested, and analyzed with nanoLC-MS/MS. **b** Mass spectra of two representative fragmented proteins ("Ras and EF-hand domain-containing protein" and "actin, cytoplasmic 2") identified in both SILAC forward and reverse. Both these fragmented proteins were found upregulated in heavy apoptotic cells (and as consequence downregulated in light live cells) and migrated at the level of gel fractions 15 (MWexp range 5–10 kDa; heavy/light ratio: 15.9134) and 16 (MWexp range 5–10 kDa; heavy/light ratio: 9.4949), respectively. The expected MWs of the entire forms of these proteins are 82.9 and 41.8 kDa, respectively (Table 1). Name, gene name, accession number, confidence of identification, sequence of identified peptide, and SILAC ratio were reported for both MS spectra. **c** Pie chart describing the CDDP-induced proteomic changes in NSCLC cells. Upregulated proteins (abundance changes >2-fold increase) are displayed in red; downregulated in blue (abundance changes <2-fold increase); no change in protein abundance in black. Fragmented proteins are showed in yellow.

highlighting a high reproducibility between the SILAC replicates (Supplementary Fig. 2a; Supplementary Data 1). Accordingly, 16 and 11 identifications resulting as fragmented proteins in forward and reverse SILAC experiments, respectively, were found upregulated in apoptotic cells (Table 1; the nature and function of these proteins are described in Supplementary Notes reported in Supplementary Informations). Among them, six fragments were identified in both SILAC experiments (Table 1). Biological process-based gene ontology (GO) analysis (PANTHER [Protein ANalysis THrough Evolutionary Relationships], version 14.1) of upregulated proteins confirmed gene enrichment categories in the frame of execution phase of apoptosis and apoptotic signaling pathway (Supplementary Fig. 2b).

To validate the relationship between proteins identified by SILAC-based proteomics and NSCLC, immunohistochemical (IHC) analyses were performed in cancer tissues derived from various patients, by using (commercially) available specific mAbs

that were obtained by immunization with the proactivator polypeptide (PSAP)$_{325–524}$, or the N terminal domain of LYRIC sequences. Both these sequences contained the peptides that were associated with the related proteins with a confidence of more than 95% by spectra analyses (Table 1; Supplementary Data 1). These mAbs detected the related antigenic proteins in both live and apoptotic NSCLC cells in tumor tissues derived from five independent patients tested, who were submitted to the surgery resection upon neo-adjuvant chemotherapy (Supplementary Fig. 3), but not in tumor tissues from two patients, who were submitted to the surgery resection without neo-adjuvant chemotherapy. These data suggest that proteins identified in freshly isolated NSCLC cells by SILAC-based proteomics are represented in NSCLC tissue from various patients after chemotherapy. As expected, anti-PSAP and anti-LYRIC antibodies detected the related proteins considerably more in live than in apoptotic tumor cells by IHC analysis (Supplementary

**Table 1 List of fragmented proteins upregulated by CDDP-treated NSCLC cells in forward and reverse SILAC experiments.**

| Accession number[a] | Protein name (gene name)[a] | Peptides[b] | Late apoptotic/ live ratio | MWcal (kDa)[a] | MWexp range (kDa) |
|---|---|---|---|---|---|
| **SILAC forward** | | | | | |
| **P53634** | **Dipeptidyl peptidase 1** (CTSC) | **6** | **3.5849** | **51,854** | **15–20** |
| Q9Y4W6 | AFG3-like protein 2 (AFG3L2) | 2 | 2.7856 | 88,584 | 15–20 |
| O75844 | CAAX prenyl protease 1 homolog (ZMPSTE24) | 2 | 100 | 54,813 | 15–20 |
| Q5VTE0 | Putative elongation factor 1-alpha-like 3 (EEF1A1P5) | 2 | 1.7272 | 50,185 | 15–20 |
| **P63261** | **Actin, cytoplasmic 2** (ACTG1) | **2** | **5.4425** | **41,793** | **5–10** |
| O95197 | Reticulon-3 (RTN3) | 2 | 2.3429 | 112,611 | 15–20 |
| Q9NQC3 | Reticulon-4 (RTN4) | 1 | 100 | 129,931 | 15–20 |
| Q92504 | Zinc transporter SLC39A7 (SLC39A7) | 1 | 11.5738 | 50,118 | 15–20 |
| P07686 | Beta-hexosaminidase subunit beta (HEXB) | 1 | 2.9101 | 63,111 | 15–20 |
| Q16891 | Mitochondrial inner membrane protein (IMMT) | 1 | 100 | 83,678 | 15–20 |
| **Q86UE4** | **Protein LYRIC** (MTGH) | **1** | **100** | **63,837** | **5–10** |
| **Q8IZ41** | **Ras and EF-hand domain-containing protein** (RASEF) | **1** | **23.1859** | **82,879** | **5–10** |
| P05141 | ADP/ATP translocase 2 (SLC25A5) | 1 | 45.9929 | 32,852 | 5–10 |
| **Q8NGV7** | **Olfactory receptor 5H2** (OR5H2) | **1** | **1.01** | **35,974** | **5–10** |
| **P07602** | **Proactivator polypeptide** (PSAP) | **1** | **3.1506** | **58,133** | **5–10** |
| P07339 | Cathepsin D (CTSD) | 1 | 2.8517 | 44,552 | 5–10 |
| **SILAC reverse** | | | | | |
| **P53634** | **Dipeptidyl peptidase 1** (CTSC) | **4** | **8.096** | **51,854** | **15–20** |
| Q9NX40 | OCIA domain-containing protein-1 (OCIAD1) | 3 | 100 | 27,626 | 5–10 |
| **P63261** | **Actin, cytoplasmic 2** (ACTG1) | **3** | **21.9962** | **41,793** | **5–10** |
| **Q8NGV7** | **Olfactory receptor 5H2** (OR5H2) | **2** | **100** | **35,974** | **5–10** |
| P12236 | ADP/ATP translocase 3 (SLC25A6) | 2 | 100 | 32,866 | 5–10 |
| **Q8IZ41** | **Ras and EF-hand domain-containing protein** (RASEF) | **1** | **1.9371** | **82,879** | **25–30** |
| Q9BQE3 | Tubulin alpha-1C chain (TUBA1C) | 1 | 100 | 49,895 | 5–10 |
| Q9P0L0 | Vesicle-associated membrane protein-associated protein A (VAPA) | 1 | 100 | 27,893 | 5–10 |
| Q9Y230 | RuvB-like 2 (RUVBL2) | 1 | 100 | 51,157 | 5–10 |
| **Q86UE4** | **Protein LYRIC** (MTDH) | **1** | **100** | **63,837** | **5–10** |
| **P07602** | **Proactivator polypeptide** (PSAP) | **1** | **18.9825** | **58,113** | **5–10** |

Protein fragments identified in both SILAC experiments were highlighted in bold.
[a]Accession number, gene name and molecular weight (MW) according to Uniprot database.
[b]Number of Unique Peptide (C.I. 95%).

Fig. 3), in agreement with the rule that antibodies generally recognize the native (folded) proteins rather than the corresponding fragmented (unfolded) forms that, vice versa, are more efficiently processed and presented to T cells by antigen-presenting cells (APCs)[31], such as those we found upregulated in apoptotic tumor cells by SILAC-based proteomics.

**T effector cell responses to multiple NM-neoAg peptides.** To assess the functional relevance of SILAC findings, we compared the immunogenicity of proteins resulted fragmented or not in CDDP-ap or live NSCLC cell line (EpT1Lu). In particular, we analyzed longitudinally memory T cell responses in peripheral blood mononuclear cells (PBMCs) isolated from 14 NSCLC (10 LUAD and 4 LUSC) patients, as compared with healthy donors (HDs), by flow cytometry (FC) (Table 2). All patients showed a stage IV NSCLC, and four of them had previously experienced a surgical treatment, without any neo-adjuvant therapy. All patients were studied before (time 0 [T0]) and after a treatment with various cycles of a chemotherapy protocol (including CDDP) (T1), whereas 12 of them were also studied after a subsequent treatment with several cycles of anti-PD-1 mAb (nivolumab) (T2) (Table 2). As a control, we also studied memory T cell responses in PBMCs from seven patients with earlier NSCLC stage (4 with IIIA, 1 with IIIB, 1 with IIA and 1 with IA), who did not require neo-adjuvant chemotherapy, and whose blood sample was obtained immediately before (1 day) the surgery resection (naive

patients) (Supplementary Table 1). The effector (eff) responses were evaluated by calculating the percentage of $CD8^+$ or $CD4^+$ T cells promptly producing interferon-γ (IFN-γ), tumor necrosis factor-α (TNF-α), or both ex vivo (i.e., without a previous stimulation in vitro), in response to a peptide matrix composed of 12 peptide pools. Each of these peptide pools contained 6–7 synthetic 20-mer peptides (overlapping of 12 residues), spanning eight randomly selected proteins that were found overrepresented in the form of fragmented proteins in CDDP-ap NSCLC cells, as compared with live NSCLC cells using SILAC-based MS analysis (i.e., olfactory receptor 5H2; Ras and EF-hand domain-containing protein; proactivator polypeptide; protein LYRIC; zinc transporter SLC39A7; ADP/ATP translocase 2; chatepsin D; ruvB-like 2) (Supplementary Fig. 4a, b). Supporting that these responses can be defined as effector memory, the majority of both $CD8^+$ and $CD4^+$ Teff cells specific to the peptide pools derived from the fragmented proteins (i.e., NM-neoAgs) were confined within the eff memory $CCR7^-CD45RA^-$ (EM) or the $CCR7^-CD45RA^+$ (EMRA) cell subsets, rather than the naive $CCR7^+CD45RA^+$ (N) or the central memory $CCR7^+CD45RA^-$ (CM) subsets, and were higher in patients than in HDs, both at T1 (after the chemotherapy protocol including CDDP) and more at T2 (after the consecutive treatment with nivolumab), but not at T0 (Figs. 2 and 3). Furthermore, the majority of responses were higher at both T1 and T2 than at T0, as well as at T2 as compared with T1, whereas the remaining

**Table 2 Demographic and clinical characteristics of NSCLC patients who experienced CDDP and subsequent nivolumab treatments.**

| | All | CDDP treatment alone | CDDP and nivolumab treatment |
|---|---|---|---|
| Number of patients | 14 | 2/14 | 12/14 |
| Gender M/F | 6/8 | 1/1 | 5/7 |
| Age (mean ± SD); years in range | 68 ± 4; 61–77 | 67 ± 3; 65–68 | 68 ± 5; 61–77 |
| Tumor histology | | | |
| LUAD | 10/14 | 2/2 | 8/12 |
| LUSC | 4/14 | 0/2 | 4/12 |
| Tumor stage | | | |
| IV | 14/14 | 2/2 | 12/12 |
| Mutations | | | |
| EGFR | 1/14 | 0/2 | 1/12 |
| ALK | 2/14 | 1/2 | 1/12 |
| KRAS | 2/14 | 0/2 | 2/12 |
| Months of overall survival (mean ± SD); range | 39 ± 21; 9–80 | 31 ± 24; 14–48 | 40 ± 21; 9–80 |
| Cycles of chemotherapy (mean); range[a] | 4.8; 3–6 | 4; 4–4 | 4.91; 3–6 |
| Cycles of Nivolumab therapy (mean); range[b] | 24; 3–84 | — | 24; 3–84 |
| Metastatic progression (instrumental validation)[c] | | | |
| Yes | 11/14 | 0/2 | 11/12 |
| No | 1/14 | 1/2 | 0/12 |
| NA | 2/14 | 1/2 | 1/12 |
| Metastatic progression (biochemical validation)[d] | | | |
| Yes | 8/14 | 1/2 | 7/12 |
| No | 3/14 | 0/2 | 3/12 |
| NA | 3/14 | 1/2 | 2/12 |

*LUAD* lung adenocarcinoma, *LUSC* lung squamous cell carcinoma, *EGFR* epidermal growth factor receptor, *ALK* anaplastic lymphoma kinase, *KRAS* Kirsten rat sarcoma virus, *NA* not tested.
[a]A single cycle of CDDP corresponds to 75 mg/m² administrated every 21 days. In this cohort of patients, CDDP was administrated in combination with one or more chemotherapeutic drugs among pemetrexed (500 mg/m²), vinorelbine (60 mg/m²), docetaxel (75 mg/m²), or gemcitabine (1000 mg/m²).
[b]A single cycle of nivolumab corresponds to 3 mg/kg administered every 14 days.
[c]Instrumental progression (after chemotherapeutic treatment) evaluated with computed axial tomography total body or positron emission tomography according to radiologic criteria of Response Evaluation Criteria in Solid Tumor 1.1.
[d]Biochemical progression (after chemotherapeutic treatment) evaluated according to increased levels of tumor biomarkers: carcinoembryonic antigen >5 ng/mL and cancer antigen 15.3 >30 U/mL.

responses tended to be, however, higher at the same times (Figs. 2 and 3). The cumulative responses (magnitude), as calculated by the means, in all patients tested, of the frequencies of CD8[+] and CD4[+] Teff cells promptly producing IFN-γ, TNF-α, or both, in response to each single peptide pools, were always higher at both T1 and T2 than at T0, as well as at T2 as compared with T1 (Fig. 4). Notably, both the magnitude and the means of several CD8[+] or CD4[+] Teff cell responses against the single NM-neoAg peptide pools were higher in the subset of naive patients showing earlier stage tumor and who did not experience any neo-adjuvant chemotherapy before surgery, as compared with HDs (Fig. 4; Supplementary Fig. 6). The antigen-specific responses were confirmed by dose–curve analyses, supporting that they recognized antigens with remarkable T cell receptor avidity (Supplementary Fig. 7). To validate the effector responses against the single peptides, we compared the percentage of patients showing CD8[+] or CD4[+] Teff cells producing IFN-γ, TNF-α, or both in response to the single peptides, at the T1 (after the chemotherapy protocol including CDDP) and T2 (after the subsequent nivolumab treatment). The resulting fold change (calculated as the ratio between percentage of responders among chemotherapy/nivolumab-treated patients, and the percentage of responders among chemotherapy-treated patients) clearly showed that the multi-specific responses were more strongly represented upon nivolumab treatment (Supplementary Figs. 8 and 9). Both CD8[+] and CD4[+] Teff cells from randomly selected patients studied rapidly degranulated, in terms of CD107a (a lysosomal-associated membrane protein-1) mobilization, in response to selected peptide pools derived from NM-neoAgs, whereas those isolated from HDs were virtually unable to perform these functions ex vivo (Supplementary Fig. 10). These effector responses were evident only upon the treatment with either the chemotherapy-alone or the consecutive nivolumab treatment. In addition, overlapping peptides derived from some random selected proteins (tubulin β-4b chain, heterogeneous nuclear ribonucleoprotein A1, glyceraldehyde-3 phosphate dehydrogenase) that were shown to be downregulated in CDDP-ap NSCLC cells, and upregulated in their live cell counterparts in the form of entire proteins, were also tested (Supplementary Fig. 11a): no significant CD8[+] or CD4[+] T cell responses against these peptides were shown in six patients analyzed, as compared with HDs (Supplementary Fig. 11b–e). Additional controls showed that the CD8[+] or CD4[+] Teff cell responses against multiple peptides derived from a known lung antigen, such as NY-ESO-1[56,57], increased upon chemotherapy, and even more upon nivolumab treatments in the peripheral blood of some patients tested, supporting the possibility that also conventional tumor antigens can be unveiled by chemotherapy (Supplementary Fig. 12).

**Caspase cleavage improves cross-presentation of NM-neoAgs.** To evaluate whether caspases have a role in preparing apoptotic tumor cell substrate for subsequent processing and presentation by APCs, we generated CD8[+] T cell lines specific to pools containing epitopes (pool 1 and pool 2) related to NM-neoAgs, and tested them for their capacity to form IFN-γ spots (in an enzyme-linked immunospot [ELISPOT] assay) within 4–6 h of contact with autologous monocytes (as APCs) that had been pulsed with: the relevant peptide pools, CDDP-ap NSCLC cells alone, CDDP-ap NSCLC cells previously treated with a selective caspase-8 inhibitor (C8I), CDDP-ap NSCLC cells previously treated with a negative caspase control (K), or lysed NSCLC cells. Both the NM-neoAg-specific CD8[+] T cell lines responded to both the direct presentation of the relevant peptide pools and the cross-presentation of CDDP-ap NSCLC cells by APCs, rather than the cross-presentation of lysed NSCLC cells (representing the proteome of live tumor cells) (Fig. 5). The treatment of CDDP-ap NSCLC cells with C8I (but not with K) blocked the cross-presentation (Fig. 5), validating the hypothesis that chemotherapy unveiled the related neoAgs by caspase cleavage in apoptotic cells. APCs that were simultaneously pulsed with C8I-treated apoptotic tumor cells and peptide pools preserved the capacity to present the latter, thus ruling out the possibility that apoptotic cells or C8I could affect the stimulatory capacities of APCs.

**Correlation of NM-neoAg-specific T cells with overall survival and PD-1 decrease.** The association between the level of functional Teff cell frequency and overall survival was studied in 12 patients, who experienced the combination of chemotherapy and immunotherapy protocol, and was computed through the log-rank (Mantel–Cox) test (Fig. 6). This analysis showed that patients' survival was correlated with the capacity of CD8[+] Teff cells, but not CD4[+] Teff cells, to produce high IFN-γ or TNF-α levels in response to NM-neoAgs (one standard deviation above the average) at T2 (following the nivolumab treatment) (Fig. 6),

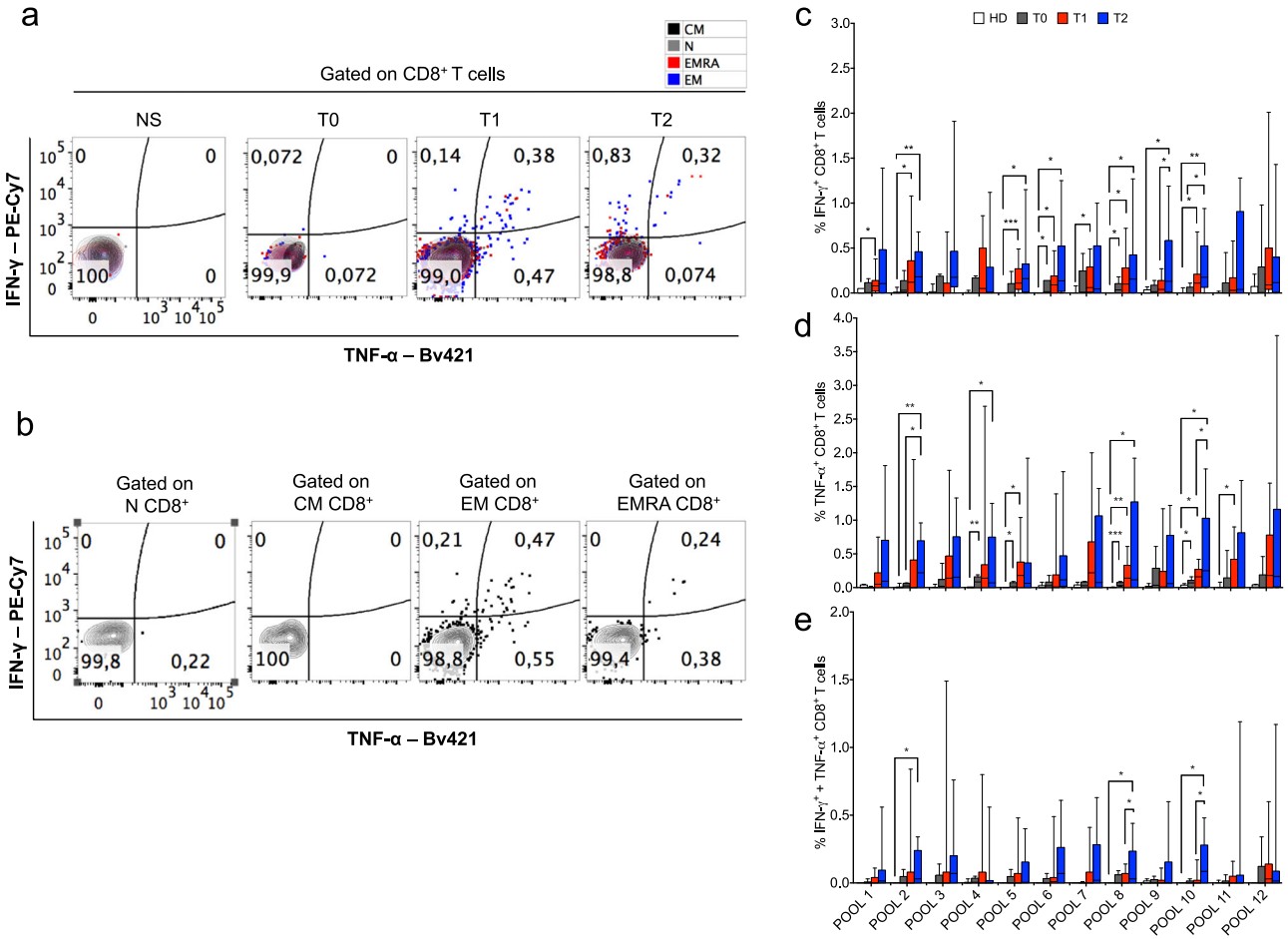

**Fig. 2 Kinetics of CD8$^+$ Teff cell responses against multiple NM-neoAg epitopes from apoptotic NSCLC cells upon chemotherapy and ICB. a** Representative FC (contour plot) analysis of cytokine production by CD8$^+$ Teff cells in response or not to the peptide pool 6 (derived from the fragmented proteins found upregulated in CDDP-ap NSCLC cells by SILAC-based MS [see Supplementary Fig. 4]) from a NSCLC patient, detected before (T0), after chemotherapy (T1), and after nivolumab cycles (T2); NS: non-stimulated cells. The naive (N) cells are indicated as gray dots, the central memory (CM) as black dots, the effector memory (EM) as blue dots, and the effector memory RA$^+$ (EMRA) as red dots. **b** Independent FC analyses of CD8$^+$ T cells producing the indicated cytokines gated within the indicated CD8$^+$ T cell subsets in a representative sample obtained at $T1$. **c–e** Percentage of the CD8$^+$ Teff cells producing IFN-γ (**c**), TNF-α (**d**), or both (**e**) in response to the 12 pools of peptides derived from the fragmented proteins upregulated in CDDP-ap NSCLC cells (as identified by SILAC-based MS). The FC analyses were performed in PBMCs obtained from 14 NSCLC patients before any treatment (T0 $N = 6$, gray bars), after the chemotherapy cycle (T1 $N = 14$, red bars), and from 12 of the 14 NSCLC patients after the nivolumab therapy (T2, blue bars), as well as in PBMCs obtained from 10 HDs (empty bars). Bars were showed as box and whisker graphs; *$p < 0.05$, **$p < 0.01$, and ***$p < 0.005$ by unpaired Student's $t$ test.

but not at T1 (after the chemotherapy protocol including CDDP). No correlation was shown between overall survival and either the number of chemotherapy or nivolumab cycles received by the single patients, or the surgery treatment previously performed in 4 out of the 12 patients, who would be submitted to the chemotherapy/immunotherapy protocol. Interestingly, the follow-up study showed that the PD-1$^+$ cells within both the NM-neoAgs-specific CD4 and CD8 Teff cells increased upon chemotherapy (T1), but then they drastically decreased upon the nivolumab therapy (T2), coming back at the start values (Fig. 7). Notably, the decrease of the PD-1$^+$ cell percentage at T2 paralleled the progressive increase of the total CD4$^+$ and CD8$^+$ Teff cells specific to NM-neoAgs (Fig. 7).

## Discussion

In this study, we identified immunogenic neoAgs generated in chemotherapy-induced apoptotic tumor cells, by combining the SILAC-based MS technology quantitatively comparing the proteome of primary CDDP-ap and live NSCLC cells, and the "T cell interrogation system," by which memory T cells longitudinally obtained from patients with late NSCLC stage, who would be submitted to CDDP and nivolumab, were used as a probe to recognize immunogenic NM-neoAgs[58]. By this multitask approach, we provided evidences showing that NM-neoAg-specific Teff cell responses increased upon chemotherapy, even more they were boosted by the nivolumab therapy, and correlated with both patients' survival and decrease of PD-1 expression.

Our data support the hypothesis that chemotherapy-induced apoptosis of tumor cells plays a pivotal role in unveiling immunogenic NM-neoAgs, as a result of caspase-dependent proteolytic activity[37,38,42]. DCs, upon phagocytosis of immunogenic apoptotic cells, have been demonstrated to efficiently process caspase-cleaved cellular fragmented proteins by proteasomes, and then to cross-present the resulting peptides on major histocompatibility complex (MHC) molecules priming T cells[37,45,48,59]. Consistent with this mechanism, here we show that memory T cells from NSCLC patients responded to epitopes derived from fragmented proteins that resulted upregulated in CDDP-ap tumor cells, as compared with their live counterparts; they did not significantly

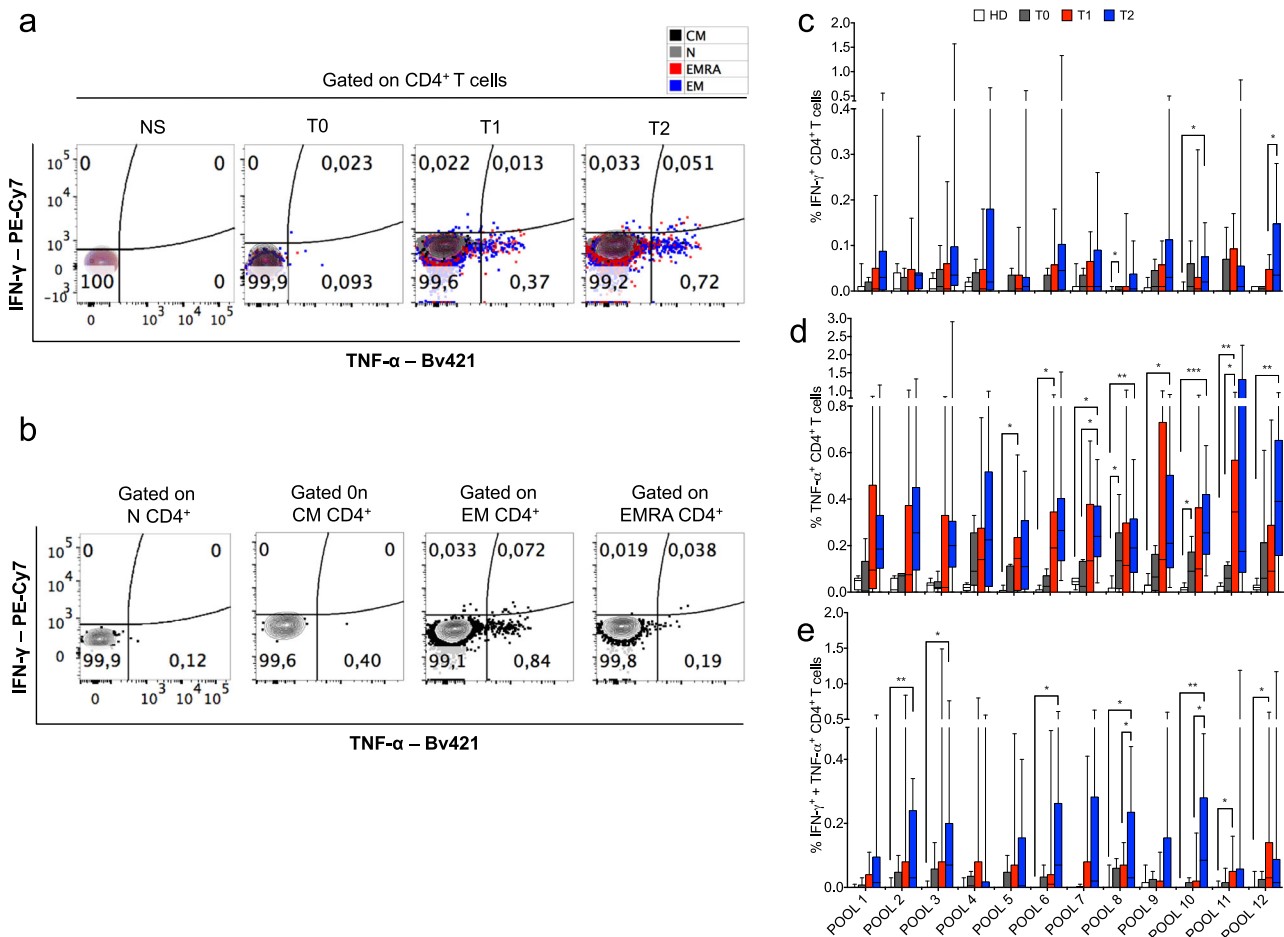

**Fig. 3 Kinetics of CD4+ Teff cell responses against multiple NM-neoAg epitopes from apoptotic NSCLC cells upon chemotherapy and ICB. a** Representative FC (contour plot) analysis of cytokine production by CD4+ Teff cells in response or not to the peptide pool 6 (derived from the fragmented proteins found upregulated in CDDP-ap NSCLC cells by SILAC-based MS [see Supplementary Fig. 4]) from a NSCLC patient, detected before (T0), after chemotherapy (T1), and after nivolumab cycles (T2); NS: non-stimulated cells. The naive (N) cells are indicated as gray dots, the central memory (CM) as black dots, the effector memory (EM) as blue dots, and the effector memory RA+ (EMRA) as red dots. **b** Independent FC analyses of CD4+ T cells producing the indicated cytokines gated within the indicated CD4+ T cell subsets in a representative sample obtained at *T*1. **c–e** Percentage of the CD4+ Teff cells producing IFN-γ (**c**), TNF-α (**d**), or both (**e**) in response to the 12 pools of peptides derived from the fragmented proteins upregulated in CDDP-ap NSCLC cells (as identified by SILAC-based MS). The FC analyses were performed in PBMCs obtained from 14 NSCLC patients before any treatment (T0 *N* = 6, gray bars), after the chemotherapy cycle (T1 *N* = 14, red bars), and from 12 of the 14 NSCLC patients after the nivolumab therapy (T2, blue bars), as well as in PBMCs obtained from 10 HDs (empty bars). Bars were shown as box and whisker graphs; *$p < 0.05$, **$p < 0.01$, and ***$p < 0.005$ by unpaired Student's *t* test.

respond to peptides derived from entire proteins resulted upregulated in live tumor cells; activation of CD8+ T cell lines specific to NM-neoAgs was obtained upon cross-presentation of apoptotic tumor cells (expressing the appropriate fragmented antigens), rather than cross-presentation of lysed tumor cells (representing the proteome of live tumor cells), and was drastically reduced by treatment of APCs with caspase inhibitors in vitro. These observations support the definition of tumor-associated NM-neoAgs as caspase-cleaved fragments that are targeted to the processing machinery and cross-presented by APCs more efficiently than their entire protein counterparts in a consistent number of patients tested, in analogy with what happens in the case of (unfolded) DRiPs as compared with the standard (folded) proteins[31]. In this context, the wide presence of effector T cell responses against these apoptosis-related fragmented proteins (as detected by ex vivo assays) in the peripheral blood from several patients upon chemotherapy suggests that these T cells, to promptly respond ex vivo, had to recently encounter the corresponding antigens in vivo, supporting the

quality of these tumor-associated fragmented proteins as public immunogenic NM-neoAgs.

It is interesting to point out that the totality of the immunogenic NM-neoAgs identified were overrepresented in CDDP-ap NSCLC cells as fragmented proteins, as compared with live NSCLC cells. This finding suggests that these self-Ags are tolerated when expressed by live cells in their complete form, but they would lose the self-identity acquiring that of NM-neoAgs, upon apoptosis generating caspase-cleaved fragmented proteins potentially more susceptible to processing and cross-presentation by APCs. We cannot exclude that other fragmented proteins can be generated in apoptotic cells by caspase cleavage that are at subthreshold level for the identification by our SILAC-based MS approach, or that diverse post-translational pathways can generate additional neoAgs.

Both NM-neoAg-specific CD8+ and CD4+ Teff cells displayed a wide range of functions including production of IFN-γ and/or TNF-α, as well as killing activity in response to the relevant epitopes ex vivo. However, the findings that CD8+ Teff cell responses

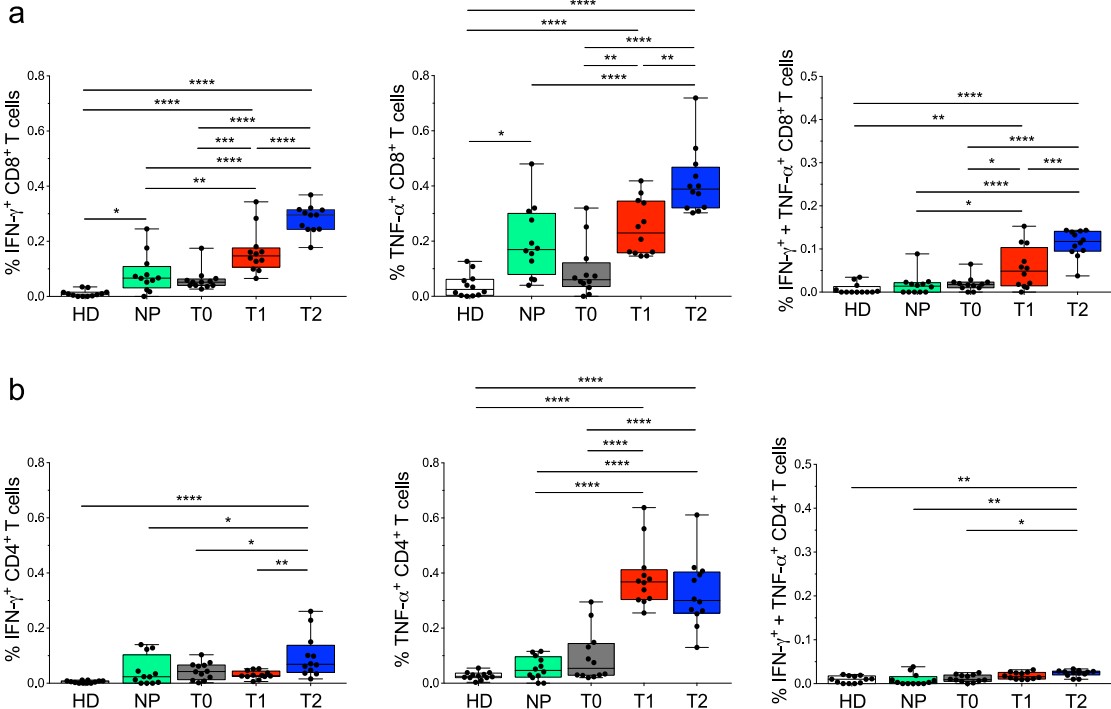

**Fig. 4 Magnitude of CD8⁺ or CD4⁺ Teff cell responses to multiple NM-neoAg epitopes. a, b** Each symbol represents the mean of percentage of CD8⁺ (**a**) or CD4⁺ (**b**) Teff cells producing IFN-γ, TNF-α, or both in response to a single peptide pool (of 12 pools) in PBMCs from HDs ($n = 10$), naive patients ($n = 7$), or in PBMCs obtained from 14 NSCLC patients before any treatment (T0 $N = 6$), after the chemotherapy cycle (T1 $N = 14$), and from 12 of the latter after the nivolumab therapy (T2). Bars were shown as box and whisker graphs; *$p < 0.05$, **$p < 0.01$, ***$p < 0.001$, and ****$p < 0.0001$ by one-way ANOVA with Tukey's multiple comparison test.

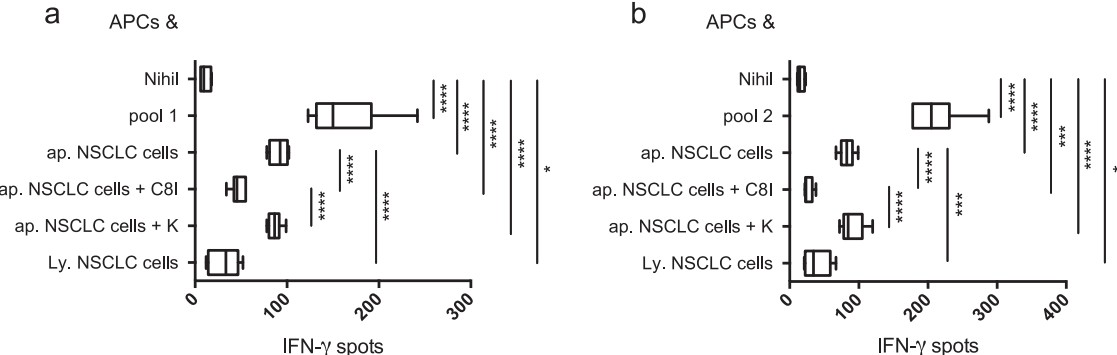

**Fig. 5 Caspase-dependent cross-presentation of NM-neoAgs from apoptotic NSCLC cells. a, b** Responses of CD8⁺ T cell lines specific to peptide pool 1 (**a**) or peptide pool 2 (**b**) derived from NM-neoAgs unveiled in CDDP-ap NSCLC cells, to autologous irradiated monocytes, as APCs, which had been pulsed or not with the related peptide pools, CDDP-ap (ap.) NSCLC cells, ap. NSCLC cells previously treated with a selective caspase-8 inhibitor (C8I), ap. NSCLC cells previously treated with a negative caspase control (K), or lysed (Ly.) NSCLC cells. Results represent the mean of two different experiments in triplicate and are expressed as IFN-γ spots in $2 \times 10^4$ lined CD8⁺ T cells. Bars were shown as box and whisker graphs; *$p < 0.05$, ***$p < 0.005$, and ****$p < 0.0001$ by unpaired Student's $t$ test.

tended to be wider than the CD4⁺ Teff responses in patients, and that only the former correlated with the overall survival suggests that CD8⁺ Teff cells provide the principal protective responses against tumors. On the other hand, we cannot rule out that CD4⁺ Teff cells can potentially contribute to mediate tumor regression[60–62]. In this context, it is not surprising that some CD4⁺ Teff cells tested displayed the capacity to degranulate in response to NM-neoAgs, since cytotoxic CD4⁺ T cells have been demonstrated to contribute to immunopathology in various forms of pathologic conditions, including cancer[60,63].

Importantly, NM-neoAgs derived from CDDP-ap tumor cells were recognized by memory T cells across several HLA-unrelated

patients ex vivo, which in turn correlated with the patients' survival. These data suggest that the resulting anti-tumor protection is likely due to the bystander effect of the wide storm of inflammatory cytokines produced by Teff cells in response to cross-presentation of apoptotic tumor cells by tissue-resident APCs, rather than by the direct killing of tumor cells, requiring an MHC-restricted antigen recognition on intact cells. In addition, these Teff cells may provide bystander (indirect) killing of neighboring live tumor cells in a non-antigen-specific manner[64]. Therefore, our system would result advantageous for the identification of public immunogenic tumor NM-neoAg epitopes recognized by multiple HLA-mismatched patients. Consistent

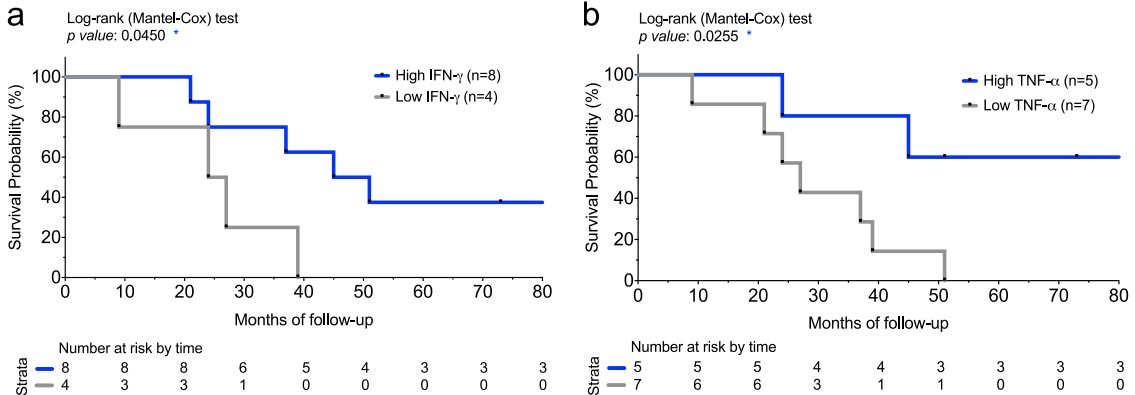

**Fig. 6 Kaplan–Meier survival curves. a, b** Each Kaplan–Meier plot presents NSCLC patients, who experienced the combination of chemotherapy and nivolumab therapy (i.e., studied at T2 $n = 12$), in two groups: "high IFN-γ (**a**) or TNF-α (**b**)" (blue) with above-average (median value) values of IFN-γ or TNF-α production in response to NM-neoAgs by CD8[+] T cells among all samples, while "low IFN-γ (**a**) or TNF-α (**b**)" (gray) with values below average. Decreased number of live patients in a time point of 10 years is reported below the curves. *$p < 0.05$ by log-rank (Mantel–Cox) test.

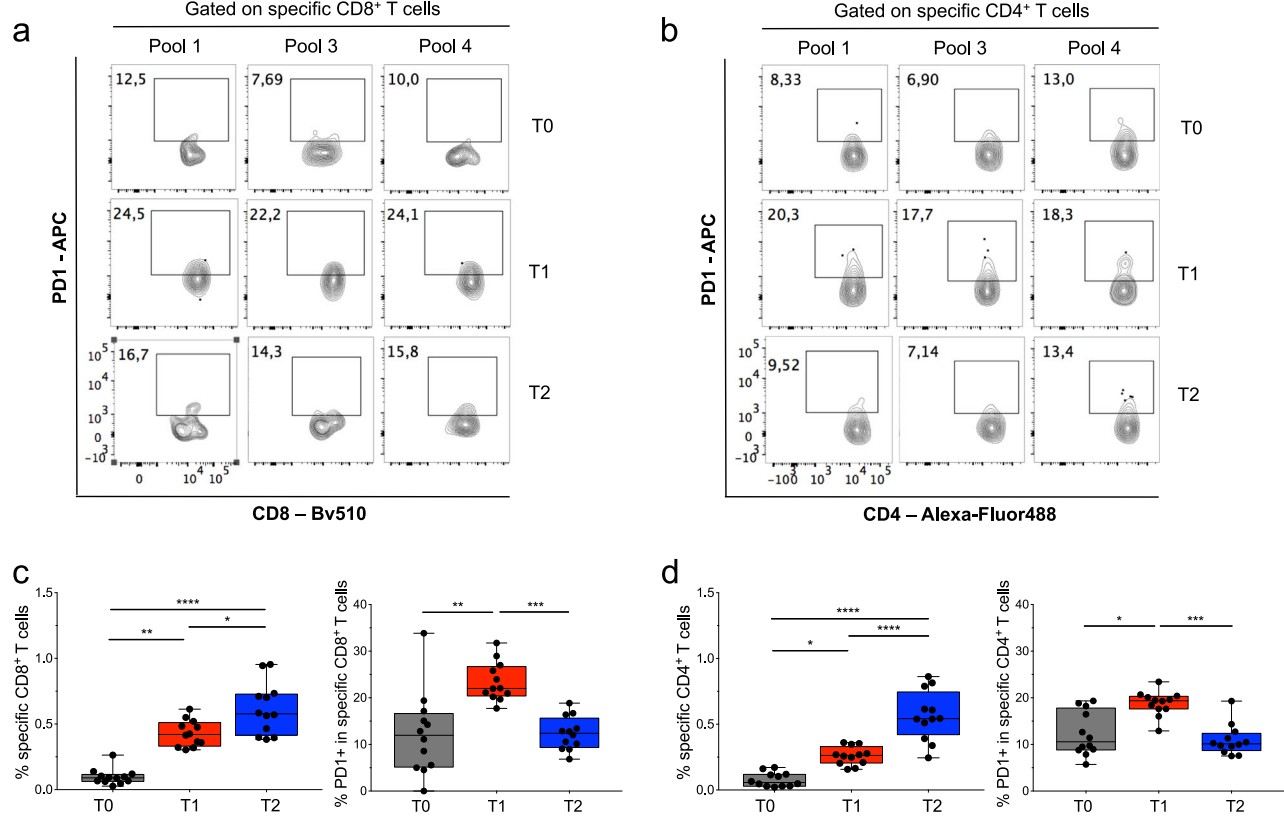

**Fig. 7 ICB therapy boosts NM-neoAg-specific CD4[+] and CD8[+] Teff cell responses in relation with a drastic reduction of PD-1 expression. a, b** Representative FC (contour plot) analysis of PD-1 expression within the total gated CD8[+] (**a**) or CD4[+] (**b**) T cells producing IFN-γ, TNF-α, and both (sum of antigen-specific CD8[+] or CD4[+] T cells) to peptide pool 1, 3, or 4. The FC analyses were performed in PBMCs obtained at T0 (before any treatment), T1 (after the chemotherapy treatment), or T2 (after the nivolumab treatment). **c, d** Each symbol in the left graphs represents the mean of percentage of CD8[+] (**c**) or CD4[+] (**d**) Teff cells producing IFN-γ, TNF-α, or both in response to a single peptide pool (of 12 pools) (here defined as specific CD8[+] or CD4[+] T cells), whereas each symbol in the right graphs represents the mean of percentage of PD-1[+] cells within the specific CD8[+] (**c**) or CD4[+] (**d**) Teff cells. The FC analyses were performed in PBMCs obtained from 14 NSCLC patients before any treatment (T0 $N = 6$), after the chemotherapy cycle (T1 $N = 14$), and from 11 patients after the nivolumab therapy (T2). Bars were showed as box and whisker graphs; *$p < 0.05$, **$p < 0.01$, ***$p < 0.001$, and ****$p < 0.0001$ by two-way ANOVA with Tukey's multiple comparison test.

with this possibility, recent evidences showed that necroptotic tumor cells confer tumor control across multiple syngeneic tumor models[54]. Although chronic inflammation can promote initial carcinogenesis by the production of several inflammatory cytokines, including IFN-γ, TNF-α, interleukin-6 (IL-6), and IL-1, in combination with the production of reactive oxygen or nitrogen species by activated myeloid cells[65–67], the same pro-inflammatory cytokines released by immune cells in an ICD milieu can favor immune activation rather than suppression in the tumor microenvironment, and enable ultimately the immune

system to act effectively against metastatic tumors[67–69]. On this basis, the beneficial effects by checkpoint inhibitors depend on the presence of a pro-inflammatory "hot" environment[70].

The finding that naive patients, who were not submitted to neo-adjuvant chemotherapy and whose blood sample was obtained the day before surgery, showed detectable frequencies of NM-neoAg-specific Teff cells in steady state, as compared with HDs, leads to the hypothesis that these responses were primed by tumor cells, made apoptotic by the natural T cell pressure against tumor progression in the early-stage tumor patients[67]. Studies on a large cohort of these patients showing pre-existing memory NM-neoAg-specific Teff cells would be of interest to verify if they could benefit from boosting ICB, upon surgery. Previous data emphasized the requirement of pre-existing memory tumor-specific T cells for unlashing the anti-tumor responses by the combination of chemotherapy and ICB or by vaccination[71,72]. The finding that these pre-existing memory T cells were almost undetectable at T0 (before chemotherapy and nivolumab treatments) in our advanced tumor patients supports the idea that they inevitably decrease in these patients, as a result of T cell exhaustion by chronic antigen stimulation[67]. Our data indicate that they can be reinvigorated following the double chemotherapy effect (i.e., increasing tumor cell apoptosis generating high levels of NM-neoAgs, and ICD induction), providing the appropriate signals (1, 2, 3, etc.) for the boosting of T cell expansion following checkpoint inhibitors, which ultimately correlates with overall survival. Whether other ICD forms, including necroptosis (occurring downstream of the receptor-interacting protein kinases RIPK1 and RIPK3)[73], can enable tumor cells to generate similar or overlapping sets of NM-neoAg repertoire that may contribute to potentiate anti-tumor immunity[54] is a crucial issue to investigate.

The observation that the percentage of NM-neoAg-specific CD4[+] and CD8[+] Teff cells expressing PD-1 increased upon chemotherapy indicates that NM-neoAgs, generated by chemotherapy-dependent apoptosis, principally activate and expand PD-1[+] tumor-specific CD4[+] and CD8[+] T cells, providing thus the substrate on which anti-PD-1 therapy can perform its boosting effect directed towards the PD-1[+] tumor-specific cell population[14,17]. This hypothesis is supported by the evidence showing that the ICB therapy induced a further increase of the frequency of NM-neoAg-specific CD4[+] and CD8[+] Teff cells that paralleled a drastic reduction of PD-1 expression and correlated with the overall survival. Consistent with this finding, PD-1 blockade has been proposed to induce a pattern of metabolic and effector T cell-specific changes that was related with an enhancement of effector molecules (e.g., IFN-γ and granzymes) and in a decrease of the exhaustion markers TIM-3 and LAG-3[14]. In addition, it has been recently demonstrated that PD-1 blockade counteracts the increase of PD-1[high] CD4[+] T follicular helper cells that accumulate at the tumor site and inhibits Teff functions in experimental tumor models[74].

Although the follow-up study of a larger cohort of patients is required to validate the association of patients' survival with CD8[+] T cell responses to the fragmented proteins identified in apoptotic tumor cells, an important facet of our study is that memory T cells targeting shared NM-neoAgs among several NSCLC patients can be detected in the peripheral blood, and can potentially be used for developing T cell-based immunotherapy across multiple cancer patients[12].

## Methods

**Study population**. Human studies were performed in accordance with the ethical guidelines of the 1975 Declaration of Helsinki and approved by the Institutional Ethical Committee (No. 2926). Informed consent was obtained from all patients. Histological diagnosis was determined based on microscopic features of carcinoma cells. We enrolled and monitored 14 stage IV NSCLC (10 LUAD and 4 LUSC) patients: all patients were studied before and after various cycles of a chemotherapy protocol (including CDDP), whereas 12 (8 LUAD and 4 LUSC) of them were also studied after a subsequent treatment with several cycles with nivolumab (Table 2), by analyzing longitudinally memory T cell responses to several peptide pools derived from proteins that had been previously identified as fragmented, or entire proteins in primary CDDP-ap or live NSCLC line by SILAC-based MS. CDDP treatment started within the first month after diagnosis in all enrolled patients. Nivolumab treatment started after failure (demonstrated by instrumental and/or biochemical metastatic progression) of CDDP chemotherapy. Demographic, clinical, and therapeutic characteristics of these NSCLC patients are described in Table 2. Possible *EGFR* and *KRAS* mutations were detected by Real-Time PCR (Applied Biosystems 7500, USA). Two out of the 14 patients could not be submitted to nivolumab after CDDP therapy, because one of them showed an *ALK* translocation and underwent oral Crizotinib target therapy, while the other patient spontaneously left our Oncology Center (Sapienza Università of Rome). As a control, T cell responses were also studied in PBMCs from seven patients with earlier stage IV NSCLCs (four with IIIA, one with IIIB, one with IIA, and one with IA), who did not require neo-adjuvant chemotherapy, and whose blood sample was obtained immediately before (1 day) the surgery resection (naive patients) (Supplementary Table 1), and in PBMCs from 10 HDs.

**Isolation and characterization of NSCLC cells**. The NSCLC cell line named EpT1Lu were established in the Immunology and Immunotherapy Unit of Regina Elena National Cancer Institute, Rome, and obtained from surgery specimen of a 69-year-old male patient, who underwent curative surgery at the Regina Elena National Cancer Institute in Rome in 2013 and who were diagnosed with a LUAD G2 T2ab N1 M0. No genomic aberration of lung cancer-specific genes *EGFR* and *KRAS* was detected in both tissue biopsy and tumor cell line by Real-Time PCR (Applied Biosystems 7500, USA) using TaqMan (CE-IVD EntroGen Kit). The patient did not receive preoperative chemotherapy or radiotherapy. Signed informed consent was obtained from patients. The present study was approved by the ethics committee of the Regina Elena National Cancer Institute in Rome (CEC/722/14). Fresh tumor tissue (within 1 h after surgical removal) was washed in phosphate-buffered saline (PBS, Corning, USA) and minced into small pieces <1 mm$^3$ using a scalpel. The specimen was enzymatically digested by trypsin (Sigma-Aldrich, USA) at a final concentration of 0.25 M for 30 min at 37 °C in a humidified incubator with 5% of $CO_2$. Then, RPMI-1640 with 10% of fetal bovine serum (FBS, Gibco, USA) was added, and after washing, cells were transferred into a standard tissue culture plate and cultured in RPMI-1640 with penicillin/streptomycin supplemented with 10% FBS and L-glutamine (Sigma-Aldrich). The cultures were maintained at 37 °C in a humidified incubator with 5% of $CO_2$. The culture medium was changed every 2–3 days. Cells were passaged after detachment when the cells reached 80–90% confluence. All the studies were performed with the initial five passages. Cell lines of epithelial origin were checked by FC staining of epithelial cell surface antigen CD326 (Ep-CAM) (Miltenyi Biotec, Germany)[75] (Supplementary Fig. 1a), as well as when cell morphology was also consistent with the epithelial origin (Supplementary Fig. 1b). The cell lines were analyzed for 24 genetic markers using the PowerPlex Fusion System Kit (BMR Genomics s.r.l., Italy).

**CDDP-dependent apoptosis of NSCLC cells**. Primary NSCLC cell lines were plated (1.5 ×10$^5$ cells/well) in 6-well plates and grown overnight. Cells were treated or not with 2-fold serial dilution of CDDP (from 1.25 to 0.078 μM). Apoptotic cells were detected by the Annexin V/Fixable Viability Dye (eBioscience, USA) and anti-active-caspase-3 (BD Biosciences, USA) staining. Each treatment was in duplicate and three different experiments were performed. After 72 h, cells were harvested, transferred into flow tubes, pelleted, re-suspended in 100 μL of fresh 1× Fixable Viability Dye eFluor780 in PBS and incubated for 30 min at room temperature. Cells were then washed and re-suspended in 100 μL of Annexin binding buffer plus Annexin V fluorescein isothiocyanate (FITC) for 15 min at room temperature. Cells were fixed and permeabilized with Cytofix/Cytoperm Kit (BD Biosciences) and stained with Brillant Violet 450 anti-active-caspase-3 for 20 min at 4 °C. After staining, samples were acquired by FC within 1 h using an LSRFortessa (BD Biosciences). Experiments were analyzed with the FlowJo software (TreeStar Inc., version 10.1r5) (Supplementary Fig. 1).

**SILAC labeling of primary NSCLC cells and sorting**. Primary NSCLC cells were grown in either SILAC heavy ($^{13}C_6$$^{15}N4$-arginine and $^{13}C_6$-lysine) or SILAC light ($^{12}C_6$$^{14}N_4$-arginine and $^{12}C_6$-lysine) conditions for eight passages before the first experiment[55]. This period lasted about 3 weeks, until the SILAC heavy cells labeling was complete (SILAC Protein Quantitation Kit, cod. 89982, Life Technologies, Thermo Fisher Scientific, USA). To analyze a reverse duplicate, NSCLC cells were also grown in SILAC light or SILAC heavy media, under the same experimental conditions. Total metabolic incorporations were verified by MS. Then, cultured NSCLC cells that were metabolically labeled with heavy isotope medium were induced to apoptosis by 0.625 μM CDDP treatment (72 h). To purify CDDP-treated apoptotic and live cells for protein identification, NSCLC cells, grown in heavy and light SILAC media, were labeled with Annexin V FITC/Fixable

Viability Dye eFluor780 as previously described. Sorting was performed using a FACSAria III (BD Biosciences) equipped with 488, 561 and 633 nm laser and the FACSDiva software (version 6.1.3; BD Biosciences). Data were analyzed using the FlowJo software (TreeStar, USA). Following isolation, an aliquot of the collected cells was evaluated for purity at the same sorter resulting in an enrichment >90% for late apoptotic and live cells (Fig. 1a). These fractions were collected, pelleted and rapidly frozen. Five experiments were performed to obtain the correct protein amount for SILAC-based MS approach. Equal amounts of protein (100 μg) of whole-cell extracts from CDDP-treated or -untreated NSCLC cell lines were mixed and separated on 4–12% gradient gels by SDS-PAGE (NuPAGE bis-tris gel, cod. NP0335box, Life Technologies, Thermo Fisher Scientific).

**NanoLC and MS analyses.** Protein digestion, peptide purification, nanoLC analysis, and MS analysis were performed as previously reported[55]. Equal protein amounts (100 μg) of whole-cell extracts from CDDP-treated or -untreated NSCLC cell lines were mixed and separated on 4–12% gradient gels by SDS-PAGE (NuPAGE bis-tris gel cod. NP0335box, Life Technologies, Thermo Fisher Scientific, USA). Gels were stained by Simply Blue Safe Stain (cod. LC6065, Life Technologies, Thermo Fisher Scientific) and 16 sections for each gel lane were cut. Protein-containing gel pieces were washed with 100 μL of 0.1 M ammonium bicarbonate (5 min at room temperature [RT]) (cod. A6141, Sigma-Aldrich). Then, 100 μL of 100% acetonitrile (ACN, cod.14261, Sigma-Aldrich) was added to each tube and incubated for 5 min at RT. The liquid was discarded, the washing step repeated once more, and the gel plugs were shrunk by adding ACN. The dried gel pieces were reconstituted with 100 μL of 10 mM DL-dithiothreitol (cod. D5545, Sigma-Aldrich)/0.1 M ammonium bicarbonate and incubated for 40 min at 56 °C for cysteine reduction. The excess liquid was then discarded and cysteines were alkylated with 100 μL of 55 mM iodoacetamide (cod. 16125, Sigma-Aldrich)/0.1 M ammonium bicarbonate (20 min at RT, in the dark). The liquid was discarded, the washing step was repeated once more, and the gel plugs were shrunk by adding ACN. The dried gel pieces were reconstituted with 12.5 ng/μL trypsin (cod. V5111, Promega Corporation, USA) in 50 mM ammonium bicarbonate and digested overnight at 37 °C. The supernatant from the digestion was saved in a fresh tube and 100 μL of 1% trifluoroacetic acid (TFA, cod. 302031 Sigma-Aldrich)/30% ACN were added on the gel pieces for an additional extraction of peptides. The extracted solution and digested mixture were then combined and vacuum centrifuged for organic component evaporation. Peptides were re-suspended with 40 μL of 2.5% ACN/0.1% TFA, desalted and filtered through a C18 microcolumn ZipTip (cod. ZTC185096, Millipore Merck, Germany), and eluted from the C18 bed using 10 μL of 80% ACN/0.1% TFA[55]. The organic component was once again removed by evaporation in a vacuum centrifuge and peptides were re-suspended in a suitable nanoLC injection volume (typically 3–10 μL) of 2.5% ACN/0.1% TFA. An UltiMate 3000 RSLC nanoLC system (Thermo Fisher Scientific) equipped with an integrated nanoflow manager and microvacuum degasser was used for peptide separation. The peptides were loaded onto a 75 μm ID NanoSeries C18 column (P/N 164534, Thermo Fisher Scientific) for multistep gradient elution (eluent A 0.05% TFA; eluent B 0.04% TFA in 80% ACN) from 5 to 20% eluent B within 10 min, from 20 to 50% eluent B within 45 min and for further 5 min from 50 to 90% eluent B with a constant flow of 0.3 μL/min. After 5 min, the eluted sample fractions were continuously diluted with 1.2 μL/min a-cyano-4-hydroxycinnamic acid (cod. C2020, Sigma-Aldrich) and spotted onto a MALDI target using an HTC-xt spotter (PAL System, Switzerland) with an interval of 20 s resulting in 168 fractions for each gel slice. MS Analysis MALDI-TOF-MS spectra were acquired using a 5800 MALDI-TOF/TOF Analyzer (Sciex, Canada). The spectra were acquired in the positive reflector mode by 20 subspectral accumulations (each consisting of 50 laser shots) in an 800–4000 mass range, focus mass 2100 Da, using a 355 nm Nb:YAG laser with a 20 kV acceleration voltage. Peak labeling was automatically done by 4000 Series Explorer software, version 4.1.0 (Sciex) without any kind of smoothing of peaks or baseline, considering only peaks that exceeded a signal-to-noise ratio of 10 (local noise window 200 m/z) and a half-maximal width of 2.9 bins. Calibration was performed using default calibration originated by five standard spots. Only MS/MS spectra of preselected peaks (out of peak pairs with a mass difference of 6.02, 10.01, 12.04, 16.03, and 20.02 Da) were integrated over 1000 laser shots in the 1 kV positive ion mode with the metastable suppressor turned on. Air at the medium gas pressure setting ($1.25 \times 10^{-6}$ Torr) was used as the collision gas in the CID off mode. After smoothing and baseline subtractions, spectra were generated automatically by the 4000 Series Explorer software. MS and MS/MS spectra were processed by the ProteinPilot software 4.5 (Sciex), which acts as an interface between the Oracle database containing raw spectra and a local copy of the MASCOT search engine (version 2.1; Matrix Science, Boston, USA). The Paragon algorithm was used with SILAC (Lys+6, Arg+10) selected as the sample type, iodoacetamide as cysteine alkylation, with the search option "biological modifications" checked, and trypsin as the selected enzyme. MS/MS protein identification was performed against the Swiss-Prot database (number of protein sequences: 254757; released on 20121210) without taxon restriction using a confidence threshold >95% (Proteinpilot Unused score ≥1.31). The monoisotopic precursor ion tolerance was set to 0.12 Da and the MS/MS ion tolerance to 0.3 Da. The minimum required peptide length was set to six amino acids. All accepted human peptides had a false discovery rate of 0.05 using reversed database searches. Quantitation was based on a two-dimensional centroid of the isotope clusters

within each SILAC pair. Ratios of the corresponding isotope forms in the SILAC pair were calculated, and lines fitting these intensity ratios gave the slope as the desired peptide ratio. The MS proteomics data have been deposited to the ProteomeXchange Consortium via the PRIDE[76] partner repository with the dataset identifier PXD016997.

**IHC analysis.** IHC analysis was performed on serial, formalin-fixed paraffin-embedded or hematoxylin–eosin-stained sections of cancer tissue samples from five NSCLC patients, who were submitted to neo-adjuvant (CDDP) therapy before surgery resection. Antibodies used were: anti-PSAP (Abcam, Cambridge, UK), anti-LYRIC/AEG1 (Abcam, Cambridge, UK) and anti-cleaved caspase-3 (Cell Signaling Technology, Danvers, MA) (Supplementary Table 2). The immunoreactions were revealed by Bond Polymer Refine Detection (Leica Biosystem, Milan, Italy) on an automated autostainer (BondTM Max, Leica). The evaluation of the different proteins was performed independently by two investigators blinded to clinical data. Apoptotic cells and apoptotic bodies were characterized in hematoxylin–eosin-stained sections by cell shrinkage, with condensed hyperchromatic nuclear chromatin and deeply eosinophilic cytoplasm, and dense extracellular or intracellular chromatin fragments, with or without associated cytoplasm, respectively. They were visualized by activated caspase-3 staining in formalin-fixed paraffin-embedded sections.

**Synthetic overlapping peptides.** Eight fragmented proteins (upregulated in CDDP-ap NSCLC cells) and three entire proteins (upregulated in live NSCLC cells) were selected for immunologic validation and analyzed for their amino acid sequence (Supplementary Figs. 4a and 11a). For each fragment a region of 20 residues before and after MS-identified sequence was selected. For custom synthesis (Mimotopes Peptide Company, Australia), 37 20-mer peptides with an overlapping region of 12 amino acids (derived from the eight fragmented proteins), as well as 22 20-mer peptides with an overlapping region of 12 amino acids (derived from the three entire proteins), in order to overcome the HLA restrictions (Supplementary Figs. 4a and 11a). Peptides were pooled into 12 mixtures of a matrix scheme[77], in which each peptide was shared between two pools and used in experiments of functional validation (Supplementary Fig. 4b). The six to seven peptides were pooled in order that each single peptide was shared between two pools according to the matrix scheme (Supplementary Fig. 4a, b). As a control, multiple overlapping NY-ESO-1 peptides (PepTivator NY-ESO-1 premium grade, human, Miltenyi Biotec) were used.

**Antigen-specific T cell assays and flow cytometry analyses.** PBMCs were isolated from fresh heparinized blood by density gradient centrifugation with lympholyte (Cedarlane, Canada), and collected in complete RPMI medium containing 10% heat-inactivated FBS (HyClone GE Healthcare Life Sciences, USA), 2 mM L-glutamine (Sigma-Aldrich), penicillin/streptomycin (EuroClone, Italy), non-essential amino acids (EuroClone), and sodium pyruvate (EuroClone). Cells were stimulated or not with 20 μg/mL peptide pool plus 1 μg/mL of anti-CD28 mAb (BD Biosciences) and the Protein Transport Inhibitor Cocktail (brefeldin A and monensin; eBioscience), or with the Cell Stimulation plus Protein Transport Inhibitor Cocktail (eBioscience) as a positive control, and mAb to CD107a for degranulation analysis, for 18 h at 37 °C. After antigen stimulation, cells were washed and stained with Fixable Viability Dye eFluor780 (eBioscience) for the exclusion of dead cells in PBS and then incubated for 30 min at room temperature. After washing, surface staining was performed by incubating cells with the labeled mAbs to CD4, CD8, CCR7, CD45RA, PD-1 and with a cocktail of labeled mAbs to CD14, CD16, CD56, CD19 (dump channel was included for the exclusion of monocytes, natural killer cells, and B cells, respectively; antibody details are reported in Supplementary Table 2) for 20 min at 4 °C in PBS containing 2% FBS. To analyze cytokine production, cells were fixed and permeabilized using the BD Cytofix/Cytoperm Fixation/Permeabilization Solution Kit (BD Biosciences) at 4 °C for 20 min, washed, and stained with mAbs to IFN-γ and TNF-α for 20 min at 4 °C in BD Perm/Wash buffer (BD Biosciences). As a control, PBMCs from some patients were also stimulated with multiple peptides derived from NY-ESO-1 (PepTivator NY-ESO-1 premium grade, human, Miltenyi Biotec), by using the same experimental procedure above. Cells were acquired with LSRFortessa cytometer (BD Biosciences) and analyzed with the FlowJo software version 10.0.8r1 (TreeStar). Gating strategy is illustrated in Supplementary Fig. 5.

**Isolation and expansion of CD8+ T cell lines.** Highly purified CD8+ T cells and monocytes were isolated from a patient's PBMCs responding to the peptide pools 1 and 2, by magnetic bead separation with the CD8+ T Cell Isolation Kit and Pan Monocyte Isolation Kit (Miltenyi Biotec), respectively. Antigen-specific CD8+ T cell lines were obtained upon repeated (bi-weekly) stimulation with autologous irradiated PBMCs that were previously pulsed with the relevant peptide pools and expanded in IL-2-conditoned medium, as previously described[37,48].

**Cross-presentation assay.** Fresh NSCLC cells ($10 \times 10^6$) were cultured in the presence or absence of 14 μg/mL C8I (ZIETD-FMK) or a negative caspase control (K, Z-FA-FMK) (BD Biosciences) for 1 h at 37 °C. Then, they were induced to undergo apoptosis by CDDP treatment and processed for validating apoptosis, as

described above. Control cells were represented by live NSCLC cells, which were promptly lysed by repeated freezing and thawing. Highly purified monocytes ($3 \times 10^4$), used as APCs, were pulsed with increasing concentrations of C8I-treated, K-treated, untreated apoptotic NSCLC cells, control lysed cells, or peptides in U-bottom 96-well plates for 18 h. Then, APCs were cultured with lined antigen-specific CD8+ T cells ($2-3 \times 10^4$), and IFN-γ spot formation by CD8+ T cells was promptly revealed after 6–8 h at 37 °C by an ELISPOT assay, as previously described[37,48]. Each peptide pool was tested in triplicate. The HLA restriction of these responses was demonstrated by blocking responses with an appropriate anti-class I mAb.

**Bioinformatic analysis of SILAC experiment**. The over-representation analysis of down- and upregulated proteins obtained from SILAC analysis was performed online with PANTHER classification system (version 14.1 – 2018_04 release; http://www.pantherdb.org). GO analysis within biological process category (cellular process [level 2] and sub-categories) was evaluated. In particular, enriched genes for "apoptotic mitochondrial changes – GO:0008637" in mitochondrion organization category, "response to toxic substance GO:0009636," and "response to drug GO:0042493" in response to chemical category were reported only in upregulated proteins (Supplementary Fig. 2B).

**Statistics and reproducibility**. Statistical analyses were performed using the software GraphPad Prism version 6.0h. To compare cytokine production in response to each single peptide pool among HD, NP, T0, T1, and T2, unpaired Student's $t$ test was used. To compare the overall cytokine production in response to all pools among HD, NP, T0, T1, and T2, one-way analysis of variance (ANOVA) test with Tukey's multiple comparison test was used. PD-1 expression analysis was performed with two-way ANOVA test. Cross-presentation and dose–response curve analyses were performed with unpaired Student's $t$ test. Kaplan–Meier survival curve with log-rank (Mantel–Cox) test was performed to compare two groups of patients selected by a median cut-off. All statistical analyses were two tailed. Significance was set at $p < 0.05$. The number of HDs or patients, whose PBMCs were submitted to immunological analyses, were described in the corresponding figure legends: when PBMCs' amount was enough, the reproducibility of flow cytometry assay was confirmed by duplicate experiments. Results obtained in the forward SILAC were found similar in the reverse SILAC, as a control, thus highlighting a high reproducibility between the SILAC replicates (Supplementary Fig. 2a; Supplementary Data 1).

**Reporting summary**. Further information on research design is available in the Nature Research Reporting Summary linked to this article.

## Data availability

All relevant data will be available from the authors upon request: A.G. (alessio.grimaldi@uniroma1.it) will be responsible for proteomics data, and I.C. (Ilenia.cammarata@uniroma1.it) for immunological data. The MS proteomics data have been deposited into the ProteomeXchange Consortium via the PRIDE[76] partner repository with the dataset identifier PXD016997. The list of proteins identified in primary NSCLC cell line by SILAC-based MS is provided in Supplementary Data 1. The source data underlying the main figures are shown in Supplementary Data 2.

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

## Acknowledgements

This work was supported by the following grants: Associazione Italiana per la Ricerca sul Cancro (AIRC) (progetti "Investigator Grant" [IG]-2014-17 i.d. 15199 and IG-2017-22 id. 19939 to V.B.; IG-2017-22 i.d. 19784 to S.P.; and IG-19822 to P.N.); the Accelerated Award 2018 (Project Id.22794 to V.B.); Ministero della Salute (Ricerca finalizzata [RF-2010-2310438 and RF-2010-2318269] to V.B.); Fondazione Italiana Sclerosi Multipla (FISM) onlus (cod. 2015/R/04 to V.B.); Fondo per gli investimenti di ricerca di base (FIRB)-2011/13 (no. RBAP10TPXK to V.B.); Istituto Pasteur Italia – Fondazione Cenci Bolognetti (grant 2014-2016 to S.P.); International Network Institut Pasteur, Paris – "Programmes Transversaux De Recherche" (PTR no. 20-16 to S.P. and V.B.). C.F. was supported by 2015 Fondazione Veronesi fellowship. We thank Rita Mancini, Claudia De Vitis (Dipartimento di Medicina Clinica e Molecolare - Sapienza Università di Roma, 00161 Rome, Italy), and Francesca Ascenzi (Dipartimento di Medicina Molecolare, Sapienza Università di Roma, 00161 Rome, Italy) for discussion and for preparing further NSCLC cell lines for future studies.

## Author contributions

A.G. performed proteomics analyses, discussed results, and contributed to the development of the study; I.C. and C.M. performed immunology experiments, discussed results, and contributed to the development of the study; C.F. collected PBMC samples, maintained primary NSCLC lines, and performed apoptosis experiments; S.P. discussed results, and contributed to the development of the study; M.B. contributed to perform immunology experiments; C.Man. performed and discussed proteomics analyses; F.L., F.B, J.R.G.B., and S.T. procured samples, recruited patients, and discussed results; M.P., R.C., and G.P.S. procured samples and discussed results; G.G. procured samples and buffy coats; G.P. performed cell sorting of apoptotic or live tumor cells; P.N. discussed results; S.S. and M.Pan. generated and characterized primary NSCLC cell lines; F.L.C. procured patients; N.D'A., F.G., and P.V. performed histologic analyses; F.F. procured patients; G.C. discussed results; V.B. conceived the study and wrote the manuscript.

## Competing interests

The authors declare no competing interests.
