## [Peer Review File · Communications Biology]

Reviewers' comments:

Reviewer #1 (Remarks to the Author):

The manuscript by Grimaldi et al reports on the progressive acquisition of T cell reactivity in peripheral lymphocytes of patients undergoing platinum-based chemotherapy + PD1-based immune checkpoint immunotherapy against specific proteolytic fragments of apoptosis-related antigens differentially expressed in a NSCLC cell line exposed in vitro to cisplatin (CDDP). The subject of this study is both topical and relevant for expanding the impact of cancer chemo-immunotherapy in that post-translational modification (PTM) of otherwise non-mutated self antigens as a consequence of platinum-based chemotherapy would indeed expand the range of tumor-specific neoantigens by which tumors can be targeted, and the data offered in support of the author's claims are of generally high-quality. Although the rules by which the phenomenon can be understood and predicted for the antigens subject to this type of PTM remain to be elucidated, I find this to be a compelling work that is likely to compel much discussion and further experimentation by which the broader applicability or the observations reported here can be more fully understood.

Reviewer #2 (Remarks to the Author):

In this paper by Grimaldi et al, the authors have used a methodology based on SILAC-based mass spectrometry to identify potential antigens in NSCLC derived from different targets following treatment with anti-cancer agents. Tumor immunotherapy is becoming an important treatment option for terminally ill cancer individuals leading to notable regression. This kind of treatment was recently approved in several countries as 1st line of treatment (together or not with chemotherapy). At its core, immunotherapy relies on the anti-tumor function of T-cells that may recognize immunogenic epitopes presented by tumor cells. Thus, in this work, the authors have attempted to develop and validate methodologies that will help better understanding from a mechanistic standpoint, how such antigens are processed and presented. In addition, this might shed light on processes linked to immunological cell-death.

Overall, the topic is of interest and the work carefully executed. The paper is well written. Still, I do have some reservations regarding certain conclusions and the methodology:

- The use of "post translationally modified" seems to me not adequate herein. Indeed, this usually refers to more defined biochemical changes (phosphorylation, sialylation, glycosylation,...).

Fragmentation of proteins (antigens) does not qualify to my humble opinion to this PTM category – after all, all the proteins are eventually degraded and I do not think honestly that PTM is the term for that kind of "modification". Thus, I strongly encourage the authors to modify the relevant terminology to prevent misleading potential readers.

- It is important to keep in mind that this work is based on the EpT1Lu cell line (a primary NSCLC line) and the question remains to what extent this represents actual processes occurring in different patients. As such, it is my opinion that it would be important to add data pertaining to another cell line (primary or known cell line) – perhaps not to redo all the analysis (as the reviewer understands the work involved) but perhaps selected experiment or basic analysis in order to be able to draw more general conclusions. It is well possible that the authors have previous (partial) data from another experimental system that may strengthen their conclusions.

- In their longitudinal analysis, the authors have analyzed the response of memory cells to synthetic peptides which would act as common tumor associated antigens – this is quite puzzling and the authors have to better explain of T-cell stimulation was performed and under which circumstances. Additional controls should be added such as multiple randomized peptides and known lung antigens (MAGE, ALK, NYESO,... see for example works by Yasumoto et al., 2009, GTC5; Mirandola et al., 2015, Oncotarget).

- In Figure 6, the authors show that patients' survival is associated with CD8+ cells to produce cytokines but it seems to the reviewer that the sample size is quite small to draw such definitive conclusions.

- Typos: "(CEC/722/14 and...)" in the methods – please correct this.

- For the degranulation assays with CD107a, it seems to me that 18hrs is too long – do the authors have data for shorter incubation time ?

Overall, this is an interesting paper summarizing a large amount of work and which should help the scientific community better understand processes involved in antigen presentation following cell death induced by different anticancer treatments.

Responses to Reviewers

Reviewer #1 (Remarks to the Author):

The manuscript by Grimaldi et al reports on the progressive acquisition of T cell reactivity in peripheral lymphocytes of patients undergoing platinum-based chemotherapy + PD1-based immune checkpoint immunotherapy against specific proteolytic fragments of apoptosis-related antigens differentially expressed in a NSCLC cell line exposed in vitro to cisplatin (CDDP). The subject of this study is both topical and relevant for expanding the impact of cancer chemo-immunotherapy in that post-translational modification (PTM) of otherwise non-mutated self antigens as a consequence of platinum-based chemotherapy would indeed expanded the range of tumor-specific neoantigens by which tumors can be targeted, and the data offered in support of the author's claims are of generally high-quality. Although the rules by which the phenomenon can be understood and predicted for the antigens subject to this type of PTM remain to be elucidated, I find this to be a compelling work that is likely to compel much discussion and further experimentation by which the broader applicability or the observations reported here can be more fully understood.

We thank so much the Reviewer for the rewarding comments and for having defined the ours a compelling work.

Reviewer #2 (Remarks to the Author):

In this paper by Grimaldi et al, the authors have used a methodology based on SILAC-based mass spectrometry to identify potential antigens in NSCLC derived from different targets following treatment with anti-cancer agents. Tumor immunotherapy is becoming an important treatment option for terminally ill cancer individuals leading to notable regression. This kind of treatment was recently approved in several countries as 1st line of treatment (together or not with chemotherapy). At its core, immunotherapy relies on the anti-tumor function of T-cells that may recognize immunogenic epitopes presented by tumor cells. Thus, in this work, the authors have attempted to develop and validate methodologies that will help better understanding from a mechanistic standpoint, how such antigens are processed and presented. In addition, this might shed light on processes linked to immunological cell-death.

Overall, the topics is of interest and the work carefully executed. The paper is well written. Still, I do have some reservations regarding certain conclusion and the methodology:

- The use of "post translationally modified" seems to me not adequate herein. Indeed, this usually refers to more defined biochemical changes (phosphorylation, sialylation, glycosylation,...). Fragmentation of proteins (antigens) does not qualify to my humble opinion to this PTM category – after all, all the proteins are eventually degraded and I do not think honestly that PTM is the term for that kind of "modification". Thus, I strongly encourage the authors to modify the relevant terminology to prevent misleading potential readers.

We thank the Reviewer for this comment, and, according to his/her suggestion, we modified the term "post-translationally modified" (PTM) to "non-mutated" (NM) neoantigens in the title and throughout the manuscript.

- It is important to keep in mind that this work is based on the EpT1Lu cell line (a primary NSCLC line) and the question remains to what extent this represents actual processes occurring in different patients. As such, it is my opinion that it would be important to add data pertaining to another cell line (primary or known cell line) – perhaps not to redo all the analysis (as the reviewer understands the work involved) but perhaps selected experiment or basic analysis in order to be able to draw more general conclusions. It is well possible that the authors have previous (partial) data from another experimental system that may strengthen their conclusions.

We agree with the Reviewer about the need to validate data obtained by SILAC-based proteomics of the EpT1Lu cell line (a primary NSCLC line) in different patients, as well as we thank the Reviewer about his/her comprehension on the work involved and on the difficulty to obtain efficient primary NSCLC lines (like EpT1Lu). According to his/her suggestion, we provided additional experimental systems in order to draw more general conclusions. So, in pags 8-9 (lines 162-181; Suppl. Figure 3), we added new experiments of immunohistochemistry showing that some of the proteins identified by proteomic analysis, were significantly expressed in cancer tissues of various patients tested. This data suggests that proteins identified in primary NSCLC cells by SILAC-based proteomics, are represented in NSCLC tissue from various patients after chemotherapy. We thank the Reviewer for suggesting us to address this issue, since the new data significantly improved the paper and further explain the presence of significant frequencies of effector T cells against them in the peripheral blood of these patients. In addition, the evidence that the fragmented rather than entire proteins were recognized by effector T cells from a wide number of NSCLC patients, is consistent with previous reports proposing that unfolded more than folded proteins can generate T cell epitopes by processing machinery (Yewdell JW et al, *Trends in immunology*, 2006). So, as a valid indirect support to the right Reviewer' comment, I would respectfully underline the general importance of our tumor associated fragmented antigens, since effector T cell responses against them, to promptly respond ex vivo (without a previous stimulation in vitro), had to recently encounter the corresponding antigens in vivo in a extensive number of patients submitted to chemotherapy.

- In their longitudinal analysis, the authors have analyzed the response of memory cells to synthetic peptides which would act as common tumor associated antigens – this is quite puzzling and the authors have to better explain of T-cell stimulation was performed and under which circumstances. Additional controls should be added such as multiple randomized peptides and known lung antigens (MAGE, ALK, NYESO,... see for example works by Yasumoto et al., 2009, GTCS; Mirandola et al., 2015, Oncotarget).

As suggested by the Reviewer, we added in the text (pag 9 line 198 and pag. 10 lines 205-210) explanation on definition antigen-specific effector T cell responses: these were defined as such due to the fact that, first, they responded to tumor peptides ex vivo (in terms of cytokine production) without a previous stimulation in vitro; second, they were predominantly confined within the EM subset. In addition, we thank again the Reviewer for suggesting the control showing that CD8+ or CD4+ Teff cell responses against multiple NY-ESO-1 peptides increased upon chemotherapy, and even more upon nivolumab treatments in the peripheral blood of some patients tested, supporting the possibility that also conventional tumor antigens can be unveiled by chemotherapy. Below we show and cumulative flow cytometry analyses in 2 patients and 4 healthy donors tested, as well as some representative analyses. We reported this data on page 11 lines 245-250 as data

not shown, but we are ready to insert a further Suppl Figure if the Reviewer and Editors should request it.

Percentage of CD4⁺ Teff cells (upper panels) or CD8⁺ Teff cells (lower panels) producing IFN- γ , TNF- α or both, in response to Peptivator NY-ESO-1-premium grade (Miltenyi). The FC analyses were performed in PBMCs obtained from 4 HDs and NSCLC patients before any treatment (T0 N=1), after CDDP-chemotherapy (T1 N=2), and after nivolumab therapy (T2 N=1).

Representative plot T1

Representative FC (contour plot) analysis of cytokine production (IFN- γ and TNF- α) by CD4⁺ Teff cells (upper panel) or CD8⁺ Teff cells (lower panel) in response or not (NS) to Peptivator NY-ESO-1-premium grade in a patient after CDDP-chemotherapy (T1). A positive control (PMA) was included.

Representative plot T2

Representative FC (contour plot) analysis of cytokine production (IFN- γ and TNF- α) by CD4⁺ Teff cells (upper panel) or CD8⁺ Teff cells (lower panel) in response or not (NS) to Peptivator NY-ESO-1-premium grade in a patient after Nivolumab therapy (T2). A positive control (PMA) was included.

- In Figure 6, the authors show that patients' survival is associated with CD8+ cells to produce cytokines but it seems to the reviewer that the sample size is quite small to draw such definitive conclusions.

We completely agree with the reviewer about the need to extend the results of survival associated with CD8 responses to a larger cohort of patients. On the other hand, these studies require several years of follow-up. However, according to R, we have underlined this need in order to draw definitive conclusions in Discussion (pag. 18; lines 420-422).

- Typos: "(CEC/722/14 and...)" in the methods – please correct this.

Thanks. We corrected by deleting "and"...

- For the degranulation assays with CD107a, it seems to me that 18hrs is too long – do the authors have data for shorter incubation time ?

Yes! We obtained similar results in degranulation assays also after 6 h (see below an example)

Overall, this is an interesting paper summarizing a large amount of work and which should help the scientific community better understand processes involved in antigen presentation following cell death induced by different anticancer treatments.

REVIEWERS' COMMENTS:

Reviewer #2 (Remarks to the Author):

My concerns have been addressed.

Responses to Reviewers

Reviewer #2 (Remarks to the Author):

My concerns have been addressed.

We thank so much the Reviewer for being satisfied with our revised paper according to his/her suggestions that have greatly improved the manuscript

.